# Marketing of medicines in primary care: An analysis of direct marketing mailings and advertisements

**Marloes Dankers**[1,2]*, **Peeter Verlegh**[3], **Karla Weber**[1], **Marjorie Nelissen-Vrancken**[1], **Liset van Dijk**[2,4], **Aukje Mantel-Teeuwisse**[5]

**1** Dutch Institute for Rational Use of Medicine, Utrecht, the Netherlands, **2** Faculty of Science and Engineering, Department of PharmacoTherapy, Groningen Research Institute of Pharmacy, Epidemiology & Economics (PTEE), University of Groningen, Groningen, the Netherlands, **3** Department of Marketing, School of Business and Economics, Vrije Universiteit Amsterdam, Amsterdam, the Netherlands, **4** Nivel, Netherlands Institute for Health Services Research, Utrecht, the Netherlands, **5** Division of Pharmacoepidemiology & Clinical Pharmacology, Utrecht Institute for Pharmaceutical Sciences (UIPS), Utrecht University, Utrecht, the Netherlands

* m.dankers@ivm.nl

**Data Availability Statement:** All relevant data for this study are publicly available from the DANS

## Abstract

### Introduction

Marketing materials from pharmaceutical companies attempt to create a positive image of marketed, often new, medicines. To gain more insight in strategies pharmaceutical companies use to influence primary care practitioners' attitudes towards marketed medicines, we investigated the use of persuasion strategies in direct marketing mailings and advertisements from pharmaceutical companies sent to general practitioners.

### Methods

General practitioners in the Netherlands were recruited to collect all direct marketing mailings, meaning all leaflets, letters and other information sent by pharmaceutical industries to the practice during one month (June 2022). Direct marketing mailings and advertisements in collected medical journals concerning medicines or diseases (together called marketing materials) were analysed according to presence of one of the seven common persuasion strategies, i.e. reciprocity, consistency/commitment, social proof, liking, authority, scarcity and unity; as well as marketed medicine and year of introduction.

### Results

Twenty general practices collected 68 unique marketing materials concerning 37 different medicines. Direct factor Xa inhibitors (n = 12), glucagon-like peptide-1 analogues (n = 5) and sodium-glucose co-transporter 2 inhibitors (n = 4) were the most frequently marketed medicines. The median year of introduction of all marketed medicines was 2012. All seven persuasion strategies were identified, with liking (64.7% of all materials) and authority (29.4%) as most prominent strategies, followed by social proof (17.6%), unity (14.7%),

repository (https://doi.org/10.17026/dans-2bj-7ach).

**Funding:** The author(s) received no specific funding for this work.

**Competing interests:** Dankers M, Weber K, Nelissen-Vrancken HJMG, and Mantel-Teeuwisse AK declare no conflict of interests. Van Dijk L received an unrestricted grant from TEVA Pharmaceuticals and Biogen for a research project not related to this study. This does not alter our adherence to PLOS ONE policies on sharing data and materials.

scarcity (13.2%), reciprocity (11.8%) and consistency/commitment (2.9%). In addition to those strategies, we identified emotional pressure (30.9%) as one commonly used new strategy.

## Conclusion

Marketing materials sent to general practices use a wide range of persuasion strategies in an attempt to influence prescription behaviour. Primary care practitioners should be aware of these mechanisms through which pharmaceutical companies try to influence their attitudes towards new medicines.

## Introduction

New medicines have been associated with increased longevity and can have benefits in terms of morbidity and health related quality of life [1,2]. However, not all new medicines have an added therapeutic value [2]. In addition, the benefit-risk ratio of new medicines has not been fully elucidated yet and new medicines are often more expensive than alternative treatments [3,4]. There is therefore an urgent need for the rational use of new medicines, both in terms of quality of care and healthcare costs, especially in the light of aging populations and rising healthcare costs [5].

In the Netherlands, primary care functions as gatekeeper of the healthcare system and plays an important role in the prescription of medicines [6]. The uptake of new medicines in primary care is often not equally distributed among physicians [7], and previous attempts to construct a universal profile of early adopters of new medicines failed [8,9]. The attitude of primary care practitioners towards new medicines is likely to play a major role in the decision to prescribe new medicines and might explain the large differences between healthcare professionals in the adoption of new medicines [10]. This attitude can be affected by a variety of factors, including marketing activities from pharmaceutical companies [11–13]. Marketing activities have been known for decades to stimulate the prescription of new medicines [14–19].

Marketing of new medicines in primary care reflects a broad set of activities, including both direct contact (e.g. medical representatives visiting the practice and educations organised by a company) and indirect contact (e.g. sponsored courses and ghost-writing) [14,19,20]. In the Netherlands, the marketing of medicines is strictly regulated and excessive inducement and dacial relations between pharmaceutical companies and healthcare professionals are prohibited [21]. Marketing of medicines therefore often happens in more subtle ways and includes the use of direct marketing mailings. Direct marketing mail is described as any marketing material that is delivered physically to a prospect's mailbox, and thus covers all kinds of paper-based marketing materials, including newsletters, flyers and brochures [22]. Another paper-based marketing activity is the use of advertisements in (medical) journals [23]. The contents of these direct marketing mailings and advertisements, further referred to as 'marketing materials', are bound to a code of conduct. This code is supervised by the Dutch Foundation for the Code for Pharmaceutical Advertising. It outlines the requirements that marketing materials must adhere to, such as providing mandatory information (e.g., about indications and adverse events) and specifies the manner in which this information is presented [21].

Although the contents of marketing materials are regulated by the Dutch code, the manner in which materials aim to influence someone's attitude towards medicines are more difficult to regulate. Influencing someone's attitude can be achieved in different ways. To explore the

**Table 1. Description of persuasion strategies by Cialdini and examples of how they can occur in pharmaceutical marketing [24,26].**

| Principle | Description | Example |
|---|---|---|
| Reciprocity | Feeling indebted to those who have helped you. | A gift from a pharmaceutical company makes healthcare professionals feeling indebted, which may lead them to change their practice in favour of the gift-giving company. |
| Consistency/ commitment | The urge to behave consistently and to commit to earlier decisions or opinions. | Agreeing to a small request (for example, a medical representative who asks a healthcare professional whether they agree that there should be more attention to disease X, or to try a new medicine on a small number of patients) increases the likelihood that the healthcare professional will start prescribing the medicine again in larger quantities. |
| Social proof | The practice of deciding what to do by looking at what others are doing. | The use of opinions of colleagues in marketing activities to sway healthcare professionals to adopt a particular therapy (e.g., 80% of your colleagues prescribe X). |
| Liking | The principle of being more likely to comply with requests made by people that are liked. | Industry representatives acting friendly towards healthcare professionals and appear to ask nothing in return, or the use of endearing pictures of patients to raise sympathy. |
| Authority | The use of individuals or institutions who are authoritative, credible and knowledgeable. | The use of key opinion leaders to convince healthcare professionals of the benefits of new medicines. |
| Scarcity | The concept that opportunities are more valuable when they are limited. | The marketing of a new medicine as 'one of a kind', or available to only a select number of practices. |
| Unity | The concept of shared identity which opens up to persuasion attempts. | A focus on cooperation and shared goals between industry and professionals will make professionals more willing to do something for the company they feel connected to. |

influence strategies used in marketing materials, we used the generally accepted framework by Cialdini [24]. This framework describes seven strategies for persuasion, that could be used to convince the recipient of the advantages of a product [24,25]. Although different taxonomies to classify persuasion strategies exist, the framework of Cialdini is widely accepted and numerous studies have shown the effectiveness of those persuasion strategies in influencing attitudes and behaviour in different areas, including pharmaceutical marketing [26–28]. This framework therefore provides a useful basis to investigate persuasion strategies in marketing materials. Table 1 provides an overview and short description of these strategies. Whether all strategies occur in direct marketing mailings and advertisements from pharmaceutical companies and whether this occurs to a similar extent is unknown.

To gain more insight in the strategies pharmaceutical companies use to influence primary care practitioners' views towards new medicines, the aim of this study was to investigate the presence and use of different persuasion strategies in marketing materials from pharmaceutical companies sent to general practitioners.

## Materials and methods

### Participant recruitment

General practitioners were recruited to collect all direct marketing mailings sent to the practice during one month. Based on a previous–non-published–pilot study, a number of 20 general practices was presumed to be enough to obtain a representative overview of direct marketing mailings. General practitioners were recruited in March and April 2022 by a call in the newsletter and social media channels of the Dutch Institute for the Rational Use of Medicine (IRUM) and members of the research team. In addition, a call was published in the Dutch Journal of Medicine ('Nederlands Tijdschrift voor Geneeskunde') [29]. Finally, symposia and conferences aimed at healthcare professionals where IRUM was represented were used to invite general practitioners.

General practitioners willing to participate were further informed about the purpose of the study and the data—including practice characteristics—that were to be collected. Practices willing to participate gave consent to participate by e-mail or telephone up to May 1, 2022. A digital confirmation of participation (by e-mail) was obtained for all practices. To investigate

the representativeness of practices, publicly available information on the practice characteristics location, number of general practitioners per practice, practice type (solo, duo or group), dispensing status and urbanisation of the location of the practice were identified by internet search.

According to Dutch legislation, approval by a medical ethics committee was not necessary, since no patients were involved in this study and the participating healthcare professionals were not exposed to interventions [30].

### Data collection

General practitioners were asked to collect all physical marketing mailings from pharmaceutical companies sent by mail from June 1 to 30, 2022. Detailed instructions were sent in the first week of May 2022. The instructions were repeated on May 31, including a final reminder to start the collection. Instructions included the collection of all direct marketing mailings, including leaflets, letters and other information sent by mail to the practice by pharmaceutical industries. Medical journals including advertisements were not to be collected, although practices were invited to collect sponsored inserts. In case of doubt, the practice was invited to collect the mailing, enabling the researches to make a selection afterwards, if necessary. A reminder to end the collection was sent by e-mail on June 30, 2022. The materials were subsequently either picked up by a member of the research team or sent to the IRUM. Practices that did not start the collection or lost their collected materials were excluded from further analysis.

### Data analysis

An overview was made of all materials received per practice by the principal investigator (MD). Multiple brochures for the same medicine in one envelop were considered as one material. Medical journals and sponsored inserts or adjusted covers were counted as separate materials. Thereafter, the selection of relevant direct marketing mailings and advertisements was made. The selection was based on two criteria regarding the sender of the mail (pharmaceutical company) and the subject (medicine or disease, to include disease awareness). All other materials were excluded.

Although medical journals were not meant to be collected and included in this analysis, we decided post hoc to include medicine advertisements in collected journals as well. This was done because of the large number of collected medical journals, despite the instruction not to do so. Moreover, advertisements in medical journals fulfilled both inclusion criteria (sender and subject) and were therefore suitable for analysis.

After inclusion and before further analysis, the marketing materials were anonymized. All unique direct marketing mailings and advertisements were classified according to the name of the marketed medicine, the medicine class (based on the Anatomical Therapeutic Chemical Classification system (ATC) 5th level) and the year of marketing approval. The year of marketing approval was based on the marketed indication. If multiple indications were marketed, the year of approval for the first indication was mentioned. The median year of approval of the medicines in all marketing materials (meaning that medicines that were marketed multiple times were also included multiple times) was calculated to gain insight in the novelty of marketed medicines. Subsequently, for every marketing material the persuasion strategies according to Cialdini's classification were captured. Prior to this analysis, a research guideline with a description of each persuasion strategy including examples from former collected marketing materials was developed and finetuned during several discussion sessions with the research team. This guideline was developed with deductive and inductive research, meaning that we analysed the pilot materials on the presence of both the strategies according to Cialdini's

classification (deductive analysis) and other strategies (inductive analysis). During this development, one additional persuasion strategy, namely emotional pressure, was identified. This additional strategy made use of the sense of responsibility or even sense of guilt of healthcare professionals, resulting in emotional pressure to do the right thing (i.e., prescribing the company's medicine). This was achieved by emphasizing the responsibility of the healthcare professional to take care of patients, often by mentioning the action the healthcare professional had to perform ("you can help her", "your patients need you to"). The strategy had some overlap with commitment, liking and unity. However, the sole focus on sense of responsibility and sense of guilt was considered as a distinct strategy to persuade the healthcare professional to prescribe the marketed medicine. After careful considerations and thorough discussions with the research team, this strategy was therefore added to the research guideline. The analysis of the collected materials was performed independently by two researchers who were primarily involved in the development of the guiding document, one with a background in pharmacy (MD) and one in marketing (PV). In addition, two other independent research assistants, one with a background in pharmacy (KW) and one in marketing (RJ), performed the analysis after being trained in using the guiding document. Cohen's kappa was calculated to measure inter-rater reliability between the two primary investigators and the investigators with the same background. Consensus was to be reached by the two primary investigators, with the opinion of a third independent researcher if needed in case of disagreement.

All results were analysed with IBM SPSS Statistics 28.0.1.1 (15).

## Results

### Baseline characteristics

Twenty-two practices signed up for the collection. Twenty out of 22 practices started and finished the collection and were thus included in the analysis and reported upon below. The characteristics of all included practices can be found in Table 2. Practices were well distributed

**Table 2. Characteristics of participating practices.**

|  | Number of practices (%) |
| --- | --- |
| Practice type |  |
| Solo | 1 (5.0) |
| Duo | 6 (30.0) |
| Group | 13 (65.0) |
| Number of general practitioners per practice |  |
| 1 | 1 (5.0) |
| 2 to 4 | 12 (60.0) |
| $\geq 5$ | 7 (35.0) |
| Dispensing status |  |
| Yes | 3 (15.0) |
| No | 17 (85.0) |
| Urbanisation level of location of practice[a] |  |
| Very strong | 4 (20.0) |
| Strong | 5 (25.0) |
| Moderate | 0 (0) |
| Little | 9 (45.0) |
| Not | 2 (10.0) |

[a]level of urbanisation is defined as very strong ($\geq 2500$ addresses/km$^2$), strong (1500–2500 addresses/km$^2$), moderate (1000–1500 addresses/km$^2$), little (500–1000 addresses/km$^2$) or not ($< 500$ addresses/km$^2$) [31].

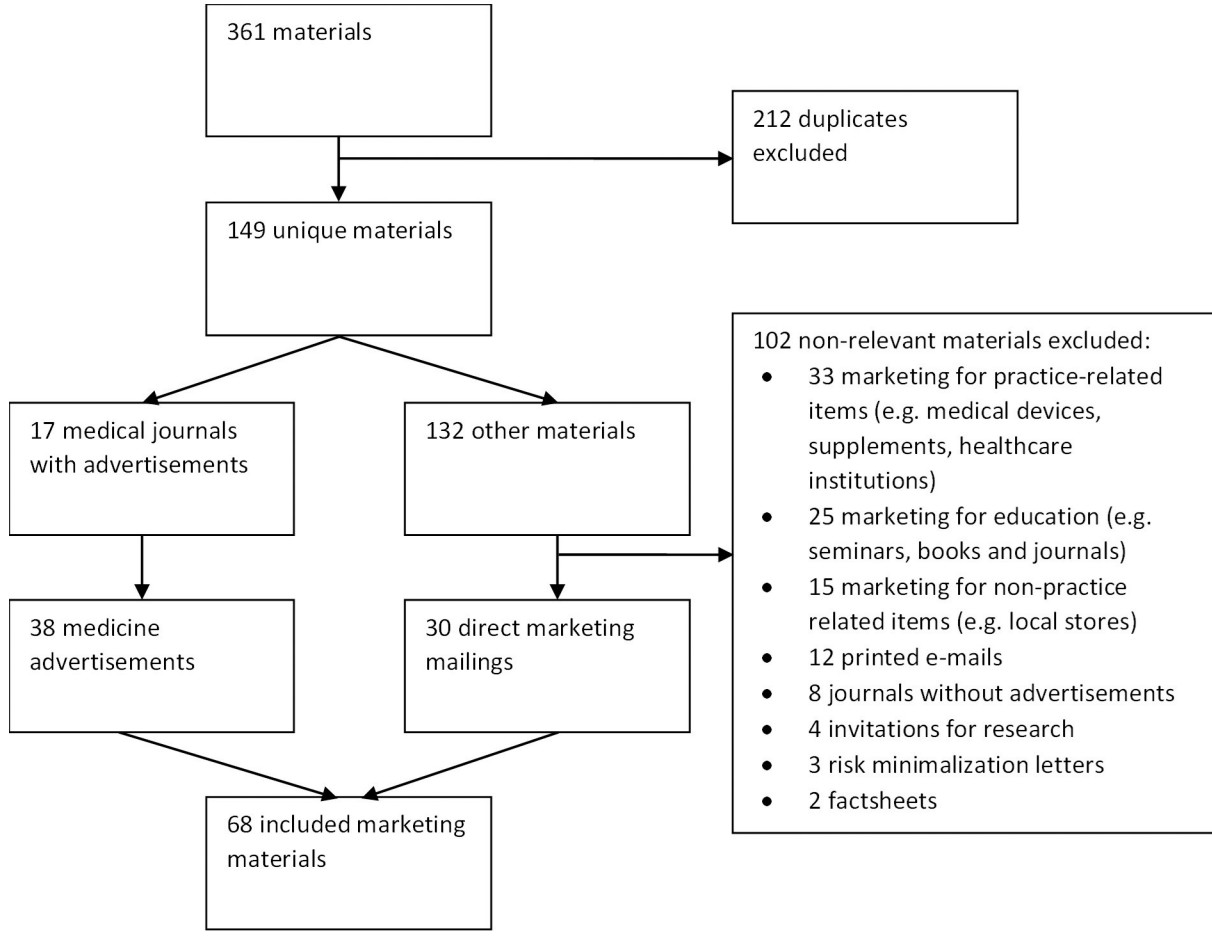

**Fig 1. Selection of marketing materials.**

across the Netherlands, 9 out of all 12 provinces were represented. The mean number of general practitioners per practice was 3.9 (range 1–11). Three participating practices were dispensing practices.

A total of 361 materials were collected (range 0–92 per practice) by the 20 included practices. One of the twenty practices only recently opened and did not receive any materials. Two other practices spontaneously reported incomplete collection, due to changing staff or inadequate communication between professionals. After removal of all duplicates, 149 unique materials (range 0–43 per practice) remained. Seventeen items were medical journals, which contained 38 unique medicine advertisements. 132 were other marketing materials, of which 30 fulfilled the inclusion criteria (range 0–14 per practice). A total number of 68 marketing materials were included for analysis (Fig 1).

## Characteristics of marketing materials

Six different types of marketing materials were identified. Advertisements in medical journals (n = 38) were most prominent, followed by marketing brochures (n = 13), sponsored inserts or covers of medical journals (n = 6) and invitations for education organised by pharmaceutical companies (n = 5). In addition, five information letters from companies about registration or reimbursement of medicines and one invitation to a company's stand with information on

**Table 3. Frequency of persuasion strategies found in collected marketing materials.**

| Persuasion strategy | Overall (n = 68) n (%[a]) | Direct marketing mailings (n = 30) n (%) | Advertisements (n = 38) n (%) |
|---|---|---|---|
| Reciprocity | 8 (11.8%) | 7 (23.3%) | 1 (2.6%) |
| Consistency/commitment | 2 (2.9%) | 2 (6.7%) | 0 (0%) |
| Social proof | 12 (17.6%) | 7 (23.2%) | 5 (13.2%) |
| Liking | 44 (64.7%) | 19 (63.3%) | 25 (65.8%) |
| Authority | 20 (29.4%) | 9 (30.0%) | 11 (28.9%) |
| Scarcity | 9 (13.2%) | 3 (10.0%) | 6 (15.8%) |
| Unity | 10 (14.7%) | 6 (20.0%) | 4 (10.5%) |
| Emotional pressure[b] | 21 (30.9%) | 9 (30.0%) | 12 (31.6%) |

[a]Percentages do not add up to 100% because multiple strategies can be used in one marketing material.
[b]Newly identified category, not described by Cialdini.

a specific disease on an upcoming medical symposium were identified. The identified materials concerned a total of 37 different marketed medicines (S1 Table). Eleven marketing materials did not mention a specific medicine. Direct factor Xa inhibitors (n = 12) were the most frequently marketed medicines, followed by glucagon-like peptide-1 (GLP-1) analogues (n = 5) and sodium-glucose co-transporter 2 (SGLT2) inhibitors (n = 4). The median year of introduction of all medicines was 2012 (range 1966–2022).

## Persuasion strategies

For the allocation of persuasion strategies, the Cohen's kappa coefficient between the two primary investigators was 0.65, indicating, according to Cohen, substantial agreement [32]. The agreement between two researchers with the same background was slightly higher (0.71 for both pharmaceutical experts and 0.80 for both marketing experts). Ultimately, agreement between the two primary researchers was reached in all cases without the need for a call from a third researcher. The frequency of identified persuasion strategies, based on consensus between the two primary investigators, can be seen in Table 3. A total of 126 persuasion strategies were found in 68 materials. No large differences existed between direct marketing mailings and advertisements. All different seven categories defined by Cialdini were identified, with liking (64.7% of all marketing materials) and authority (29.4%) as the most represented persuasion strategies. We identified emotional pressure as an additional category, which was present in 30.9% of all materials.

**Reciprocity.** Reciprocity refers to the obligation to help those who have helped you and is often expressed by providing someone with something that could be considered as a gift. In the marketing materials, examples of reciprocity were found eight times. For example, some invitations for sponsored educations advertised free meals and accreditation points. Other examples were offering free samples, books and placebo-inhalers. Most identified gifts were relatively small. The gifts were both aimed at the practice as a whole (for example training inhalers) or at individual general practitioners (for example accreditation points).

Invitation for a sponsored education with free meals and accreditation points.

*#Marketing material 37, invitation for education.*

**Consistency/Commitment.** Consistency refers to behaving consistently and to commit to earlier decisions or opinions. In marketing materials, this can be achieved by the use of (semi-) rhetorical questions. A positive answer on these often obligate questions automatically implies that the marketed medicine is the best option. This mechanism was identified two times in the collected materials.

> "Do you and your patients prefer ease of use and ease of prescription?"
>
> *#Marketing material 24, brochure.*

**Social proof.** Social proof is the use of opinions of colleagues to promote a product. In marketing materials, this can be achieved by using opinions or actions of other healthcare professionals. A referral to a healthcare professional who is positioned as an expert in the field, was considered as authority and not social proof.

In the collected materials, we found several marketing materials stating 'the most prescribed medicine for disease X'. Remarkably, in a specific therapeutic class, the statement of being the most prescribed medicine was found for two different medicines, referring to different investigations. Social proof was also used more subtly by the use of specific pictures referring to other physicians' actions, for example by using white coats or stethoscopes.

> "Most prescribed [medicine class X] in the Netherlands."
>
> *#Marketing material 4, brochure.*

**Liking.** Liking is the creation of a positive feeling about a company or product. Liking was the most identified persuasion strategy and used in almost two-thirds of all marketing materials. Liking was most often achieved by the use of sympathetic pictures, for example of friendly-looking patients, beautiful landscapes and animals. Liking was achieved by portraying patients as sad people who could be helped on one hand and as self-confident people who had already been helped by the product on the other hand.

> A portrait of a happy-looking boy playing the guitar accompanied with the phrase "Be who you want to be"
>
> *#Marketing material 59, advertisement.*

**Authority.** Authority refers to the use of individuals or institutions who are authoritative, credible and knowledgeable. In addition, authority can also be created by focusing on the authority of the product itself, by emphasizing the status of the medicine. Authority was identified twenty times and attained by mentioning the authority of the pharmaceutical company as well as the use of authority of others to emphasize the medicines' benefit. Authority of the company was for example emphasized by mentioning the years of experience in a specific field. Materials also referred to the authority of others, for example by referring to guidelines, official institutions like registration authorities and professional organisations, and individual medical experts. Authority was also achieved by focussing on the seniority of the product.

> "[Medicine X] has been a reliable [medicine group X] for almost 50 years and has been used by 3 million Dutch women."
>
> *#Marketing material 8*, *sponsored cover*.

**Scarcity.** Scarcity refers to limited options that are considered more valuable. In marketing materials, this can be achieved by focusing on the unique status of a product. The collected marketing materials made use of this scarcity by referring to a medicine as 'the only one'. Often, the phrase 'the first and only' was used. Emphasized characteristics referred among others to indications, dosage forms and mechanisms of action.

> "[Medicine X] is the first and so far only selective [medicine group X] registered for the aforementioned indication."
>
> *#Marketing material 19*, *information letter*.

**Unity.** Unity is the principle of shared identity, which can be explained as shared identity between the producer and healthcare professionals. In marketing materials, this was expressed by positioning the company next to the professional, to emphasize that they were on the same side and had the same goals. This was often done by using the word 'together', but also by phrases like 'we can help you' and statements implying that the company was helping the healthcare professional by providing them with therapeutic options for their patients.

> "Together we tackle overweight".
>
> *#Marketing material 38*, *invitation for education*.

**Emotional pressure.** In addition to the persuasion strategies defined by Cialdini, one more strategy was identified in the materials, which was the second most common, after liking. This strategy made use of the sense of responsibility or even sense of guilt of healthcare

> "Provide your patients with [disease X] and ['old' medicine X] with better chances with ['new' medicine Y]."
>
> *#Marketing material 79*, *advertisement in magazine*.
>
> An image of a granddaughter hugging her grandfather, accompanied with the phrase: "838 additional hugs from grandfather, due to the protection you provide your [disease X] patients with."
>
> *#Marketing material 15*, *sponsored cover*.
>
> "For which T2DM patient do you want to do more?"
>
> *#Marketing material 13*, *brochure*.

professionals. This strategy was often achieved by addressing healthcare professionals to do what was best for their patients (i.e. prescribing the company's medicine).

## Discussion

Marketing materials sent by pharmaceutical companies to general practitioners used a wide range of persuasion strategies, of which liking and authority were the most common. All other persuasion strategies defined by Cialdini [24], i.e. reciprocity, consistency/commitment, social proof, scarcity and unity, were also used, often in combinations. In addition to these strategies, one additional category, coined 'emotional pressure', was identified. The presence of eight different persuasion strategies in 68 marketing materials indicates that pharmaceutical companies use a wide range of strategies to influence the attitudes of healthcare professionals towards prescribing their (new) medicines.

The identified persuasion strategies were achieved by use of text and images and often a combination of these. Persuasion strategies in marketing materials were identified on different levels. Although most materials were aimed at creating a positive image of a specific medicine, marketing materials also focussed on disease awareness or positively portraying the company itself. The identified persuasion strategies have been associated with different motives of persuaders. Reciprocity, liking and unity have been primarily associated with cultivation of a relationship. Social proof and authority are often used to reduce uncertainty, and consistency and scarcity are regularly involved if call to action is the primary goal [24]. The presence of all these different persuasion strategies in the marketing materials implies that all goals are being pursued. However, with liking and authority as the most prominent Cialdini strategies [24], it can be argued that building a relationship and reducing uncertainty are the most prominent goals of the investigated materials.

In addition to the persuasion strategies described by Cialdini [24], we identified one additional strategy, which was described as emotional pressure. This strategy makes use of the sense of responsibility of healthcare professionals, implying that the prescription of the marketed medicine is the best care they can provide for their patients. The importance of emotion in persuasion has been recognized before [33,34]. In consumer research, a similar tactic of using emotions to elicit feelings of accountability and responsibility has been studied in the promotion of socially responsible products and behaviours [35]. This newly identified strategy in addition to the model of Cialdini most probably reflects the unique situation of medicine marketing, where the choice for a specific product is made by a professional, rather than by a consumer. Since the Cialdini principles are not exclusively developed for medicine marketing, this might explain why this principle based on professional attitude was identified in the collected marketing materials, but not described in Cialdini's framework.

The presence of persuasion strategies in marketing activities of pharmaceutical companies has been described before [26]. However, to the best of our knowledge, this is the first study to investigate the use of the persuasion described by Cialdini in marketing materials from pharmaceutical companies. In previous research, direct marketing brochures and advertisements have been shown to have little or no educational value [36–38]. In addition, studies have also shown that marketing materials contain inaccurate or even misleading statements [23,36–39]. Although marketing activities have been shown to directly influence prescription behaviour [14–18,20,40,41], healthcare professionals still underestimate their vulnerability to marketing, thinking they themselves are not affected by marketing activities [14,18,24]. In the advertising literature, this is known as the third person effect, the illusion that advertising influences other people but not me [42]. The strategies used however have been proven to influence behaviour, even if the recipient is not aware of this. The crux of these persuasion strategies is that they

produce a distinct kind of automatic, mindless compliance [24]. The ultimate effect of the identified persuasion strategies, also in relation to other marketing activities such as visits by medical representatives and sponsored educations, was not investigated in this study and calls for further research. However, because of the proven efficacy of these persuasion strategies [24,25], and the proven impact of other marketing activities by pharmaceutical companies on prescription behaviour [14–18,20,40,43], there is no reason to believe that the marketing materials would not influence healthcare professionals. The lack of educational value and the wide presence of persuasion strategies makes it even more clear that direct marketing mails and advertisements should be viewed as promotional information and emphasizes the urge to create awareness of the mechanisms marketing materials use to influence decision-making.

Our study focussed solely on marketing materials sent to general practices. The decision to include advertisements in addition to direct marketing mailings was made post hoc. Since the identified persuasion strategies in direct marketing mailings and advertisements did not really differ, the decision to include both seems justified. Marketing brochures and advertisements from other sources–for example symposia, sales representatives or medical journals not collected by the general practices–were not included. In addition, other marketing activities such as digital marketing and indirect marketing were not assessed in this study. Different marketing activities from pharmaceutical companies are known to reinforce each other and have a synergistic effect on prescription behaviour [14,41]. It is therefore important to realise that the persuasion strategies identified in this study are only a small part of all attempts to influence prescription behaviour. Although marketing materials are only a small part of all marketing activities, it has been present since for decades. Already in 1939, the amount and effect of direct marketing mailings towards healthcare professionals were investigated. At that time, the average number of advertising mail per healthcare professional was approximately four pieces per day [18]. In our study, the number of unique marketing materials per general practice in four weeks' time ranged from 0 to 43. It is not known whether the wide range of received materials reflects a real difference in the extent of that pharmaceutical companies target general practices, or a difference between practices in the adherence to collection instructions. A number of practices spontaneously reported incomplete or inadequate collection, indicating that no firm conclusions about the number of marketing materials could be made. The number of collected marketing materials in our study indicate that marketing materials should still be seen as a relevant element of all marketing activities.

A wide range of introduction years of the marketed medicines was identified in this study. The novelty of the marketed medicines was less than anticipated, with 2012 as median introduction year and 1966 as first introduction year. The wide range of introduction years is probably related to the relatively slow uptake of new medicines in Dutch general practices [44,45], explaining why pharmaceutical companies continue marketing activities years after the launch of their product. It also points out the importance of alertness to marketing activities, even if medicines are not considered to be new anymore.

The main strength of this study is the large number of included marketing materials – obtained from brochures and medical journals collected by general practices – and the focus on persuasion strategies in text and image, resulting in a clear overview of how medicines are marketed in direct marketing mails and advertisements by pharmaceutical companies. There are however also some limitations. First, it is not known whether the included marketing materials were representative for all marketing materials, since we included only marketing materials sent to a limited number of practices during one month and it is not known whether all practices followed the exact instructions. In addition, the decision to include advertisements as well was made post hoc, indicating that the included advertisements did not reflect the total number of advertisements in this month. However, although the marketed medicines are likely

to be time-dependent, the identification of all different persuasion strategies makes it unlikely that the conclusions about the use of persuasion strategies would significantly alter when including other marketing materials. Second, the allocation of persuasion strategies to marketing materials can be subjective. However, the interrater analysis showed a substantial agreement between the different assessors and the use of two researchers with marketing expertise and pharmaceutical expertise minimised this risk.

This study provides a clear overview of marketing materials sent to general practices in June 2022 and sheds light on the used persuasion strategies. Primary care practitioners should be aware of these mechanisms used by companies, to ensure that they are as little as possible influenced by this kind of marketing. Furthermore, practitioners should be educated in recognizing and countering these kind of persuasion strategies in order to prevent unwanted influence [46]. Training in resistance strategies [47] may provide a valid starting point.

## Supporting information

**S1 Table. Characteristics of marketed medicines.** IUD: Intrauterine device. [a]the year of registration refers to the year of registration for the marketed indication. If multiple indications were marketed, the first year was mentioned. [b]the marketing materials for allergen extracts were for different patented products and have therefore different years of introduction. [c]the marketing materials for levonorgestrel were for different dosage forms and have therefore different years of introduction. [d]the marketing materials for testosterone were for different dosages and dosage forms and have therefore different years of introduction.
(DOCX)

## Acknowledgments

The authors would like the general practitioners for the collection of marketing materials and Roos Janssen for the evaluation of persuasion strategies in the materials.

## Author Contributions

**Conceptualization:** Marloes Dankers, Marjorie Nelissen-Vrancken, Liset van Dijk, Aukje Mantel-Teeuwisse.

**Data curation:** Marloes Dankers.

**Formal analysis:** Marloes Dankers, Peeter Verlegh, Karla Weber.

**Investigation:** Marloes Dankers, Peeter Verlegh, Karla Weber.

**Methodology:** Marloes Dankers, Peeter Verlegh, Marjorie Nelissen-Vrancken, Liset van Dijk, Aukje Mantel-Teeuwisse.

**Project administration:** Marloes Dankers.

**Supervision:** Marjorie Nelissen-Vrancken, Liset van Dijk, Aukje Mantel-Teeuwisse.

**Validation:** Marjorie Nelissen-Vrancken.

**Visualization:** Marloes Dankers.

**Writing – original draft:** Marloes Dankers.

**Writing – review & editing:** Peeter Verlegh, Karla Weber, Marjorie Nelissen-Vrancken, Liset van Dijk, Aukje Mantel-Teeuwisse.

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
