## [Decision Letter · Decision Letter 0]

28 Apr 2023

PONE-D-23-07120Marketing of new medicines in primary care: an analysis of direct marketing mailings and advertisementsPLOS ONE

Dear Dr. Dankers,

Thank you for submitting your manuscript to PLOS ONE. After careful consideration, we feel that it has merit but does not fully meet PLOS ONE’s publication criteria as it currently stands. Therefore, we invite you to submit a revised version of the manuscript that addresses the points raised during the review process.

am writing to inform you of my decision regarding the manuscript under consideration. After careful evaluation of the manuscript, I have decided to grant it a status of "major revisions."

While the manuscript shows promise, there are several areas that require substantial improvements before it can be considered for publication. In particular, I have identified several major flaws in the methodology used, and the authors need to provide more rigorous analysis and interpretation of their results. Additionally, the writing needs significant refinement, particularly in terms of clarity and organization.

I believe that with significant revisions and improvements, the manuscript has the potential to make a valuable contribution to the field. Therefore, I would like to invite the authors to revise and resubmit their manuscript.

Thank you for your attention to this matter. Please let me know if you have any questions or concerns.

Sincerely,

We look forward to receiving your revised manuscript.

Kind regards,

Naeem Hasan Mubarak, PhD

Academic Editor

PLOS ONE

“I have read the journal's policy and the authors of this manuscript have the following competing interests: Dankers M, Weber K, Nelissen-Vrancken HJMG, and Mantel-Teeuwisse AK declare no conflict of interests. Van Dijk L received an unrestricted grant from TEVA Pharmaceuticals and Biogen for a research project not related to this study.  “

Additional Editor Comments:

I am writing to inform you of my decision regarding the manuscript under consideration. After careful evaluation of the manuscript, I have decided to grant it a status of "major revisions."

While the manuscript shows promise, there are several areas that require substantial improvements before it can be considered for publication. In particular, I have identified several major flaws in the methodology used, and the authors need to provide more rigorous analysis and interpretation of their results. Additionally, the writing needs significant refinement, particularly in terms of clarity and organization in the introduction and discussion section.

I believe that with significant revisions and improvements, the manuscript has the potential to make a valuable contribution to the field. Therefore, I would like to invite the authors to revise and resubmit their manuscript.

Thank you for your attention to this matter. Please let me know if you have any questions or concerns.

Sincerely,

Reviewers' comments:

Reviewer's Responses to Questions

**Comments to the Author**

1. Is the manuscript technically sound, and do the data support the conclusions?

Reviewer #1: Yes

Reviewer #2: Yes

2. Has the statistical analysis been performed appropriately and rigorously? 

Reviewer #1: N/A

Reviewer #2: Yes

3. Have the authors made all data underlying the findings in their manuscript fully available?

Reviewer #1: Yes

Reviewer #2: Yes

4. Is the manuscript presented in an intelligible fashion and written in standard English?

Reviewer #1: Yes

Reviewer #2: Yes

5. Review Comments to the Author

Reviewer #1: This article evaluates persuasion strategies used in mailed print advertisements in the Netherlands to point out how prevalent they are and their potential impact on prescribing behaviour. The focus on persuasion strategies is very important because as the authors point out, physicians are largely unaware of how vulnerable they are to this type of promotional tactic. However, the authors treat mailed material in isolation from other types of promotion whereas they operate synergistically to reinforce each other. The authors acknowledge that they only collected one type of promotion but in their discussion, they need to talk about the synergistic effect of multiple types of promotion each reinforcing the other.

The authors should give more detail about how the additional strategy of emotional pressure was identified during the course of their study.

The authors should explain what is (and is not) allowed in written promotional material in the Netherlands, i.e., based on either industry codes and/or government codes.

My understanding is that print promotional material is less important to pharmaceutical companies than it use to be. Do the authors have an estimate of how important direct marketing mail is as a proportion of all promotion in the Netherlands, e.g., in terms of the percent of all promotion expenses or in return on investment?

Did the authors include material left behind by company sales representatives who visited doctors' offices?

Were the medical journals that general practitioners collected ones that they subscribed to or ones that were sent to them for free? Was there any difference in the type of persuasion strategy between the ads in the two different types of journals?

In investigating the representativeness of the practices, did the authors collect data about whether general practitioners saw sales representatives and if so how often, e.g., visits per week? Seeing sales representatives may be a proxy for the influence that direct mailing might have on prescribing behaviour.

In discussing educating practitioners about persuasion strategies the authors might want to refer to the following article: Mansfield et al. Educating health professionals about drug and device promotion: advocates’ recommendations. PLoS Medicine 2006;3:e451.

Reviewer #2: 1. Median year of launch of studied medicine was 2009 and the study conducted in June 2022. So, the word “new” may be omitted from the title further more in line 401 – 405 authors have mentioned that there is slow uptake of new medicine in Dutch practices.

2. Why the authors have calculated median year of launch? Justify your argument.

3. Provide of list of medicine with launch year.

4. 1966- first year of introduction, such products are called cash cow products which generally need no marketing so authors should provide pictures of marketing material of such products.

5. Dr. Cialdini is very well known for his work on “science of influence” but his work has also been criticized globally, so I suggest that authors shall give more references on the seven attributes of influence.

6. Emotional pressure is the eighth attribute for which references are required.

7. Authors should amend legend of Table I because in the legend they have stated seven strategies of influence by Cialdini but they have actually mentioned 8 strategies.

8. Authors have cited 41 articles out of which 31 references are older than 5 years so much so that one reference is from an article published in 1940. If use of old references is in conformity with journal's policy then it is ok.

9. Reference number 29 is on “urbanization data” and should be with more detail in the reference list.

6. PLOS authors have the option to publish the peer review history of their article (what does this mean?). If published, this will include your full peer review and any attached files.

Reviewer #1: **Yes: **Joel Lexchin

Reviewer #2: No

---

## [Author Response · Author response to Decision Letter 0]

14 Jun 2023

Utrecht, 9 June 2023

Dear editor,

We would like to thank the reviewers for their useful comments on our manuscript ‘Marketing of new medicines in primary care: an analysis of direct marketing mailings and advertisements’ (Manuscript ID: PONE-D-23-07120). In response to their comments, we have revised the manuscript. In particular, we provided more information about the methodology of our study, the rationale for investigating marketing materials only and the use of the framework of Cialdini. 

We also like to inform the editor that we discovered a calculation error in the median year of introduction of the included medicines during the revision of our manuscript. We apologize for this mistake and are thankful for the opportunity to correct this error at this stage. We would like to reassure you that this correction does not affect the overall conclusions and implications of our study. 

Attached to this letter, you will find a detailed response to the comments of the reviewers. 

We hope the manuscript will now be suitable for publication in PlosONE.

Yours sincerely,

Marloes Dankers, MSc

 

Additional Editor Comments:

I am writing to inform you of my decision regarding the manuscript under consideration. After careful evaluation of the manuscript, I have decided to grant it a status of "major revisions."

While the manuscript shows promise, there are several areas that require substantial improvements before it can be considered for publication. In particular, I have identified several major flaws in the methodology used, and the authors need to provide more rigorous analysis and interpretation of their results. Additionally, the writing needs significant refinement, particularly in terms of clarity and organization in the introduction and discussion section.

I believe that with significant revisions and improvements, the manuscript has the potential to make a valuable contribution to the field. Therefore, I would like to invite the authors to revise and resubmit their manuscript.

Thank you for your attention to this matter. Please let me know if you have any questions or concerns.

Sincerely,

We thank the editor and both reviewers for their useful feedback. For a detailed description of all amendments, please see below. 

 

Reviewers' comments:

Reviewer's Responses to Questions

Comments to the Author

1. Is the manuscript technically sound, and do the data support the conclusions?

Reviewer #1: Yes

Reviewer #2: Yes

2. Has the statistical analysis been performed appropriately and rigorously? 

Reviewer #1: N/A

Reviewer #2: Yes

3. Have the authors made all data underlying the findings in their manuscript fully available?

Reviewer #1: Yes

Reviewer #2: Yes

4. Is the manuscript presented in an intelligible fashion and written in standard English?

Reviewer #1: Yes

Reviewer #2: Yes

5. Review Comments to the Author

Reviewer #1: This article evaluates persuasion strategies used in mailed print advertisements in the Netherlands to point out how prevalent they are and their potential impact on prescribing behaviour. The focus on persuasion strategies is very important because as the authors point out, physicians are largely unaware of how vulnerable they are to this type of promotional tactic. However, the authors treat mailed material in isolation from other types of promotion whereas they operate synergistically to reinforce each other. The authors acknowledge that they only collected one type of promotion but in their discussion, they need to talk about the synergistic effect of multiple types of promotion each reinforcing the other.

We thank the reviewer for this acknowledgement of the importance of the subject of our study. In response to the feedback, we amended the discussion and emphasized the reinforcement of different marketing activities (line 428). 

The authors should give more detail about how the additional strategy of emotional pressure was identified during the course of their study.

The additional strategy was identified during the development of the research guideline. After careful consideration in the research team, it was distinguished as an additional strategy and added to the research guideline. We describe this process now in more detail in the methods (line 188). In addition, we amended the discussion by providing a possible explanation for the lack of this principle in Cialdini’s model (line 390) and made a reference to previous studies on the role of emotion in persuasion and marketing (line 384) 

The authors should explain what is (and is not) allowed in written promotional material in the Netherlands, i.e., based on either industry codes and/or government codes.

In the introduction, we have now added a description of the Dutch Code of Conduct for Pharmaceutical Advertising which offers guidelines for what is and is not allowed in written promotional material (line 90). 

My understanding is that print promotional material is less important to pharmaceutical companies than it use to be. Do the authors have an estimate of how important direct marketing mail is as a proportion of all promotion in the Netherlands, e.g., in terms of the percent of all promotion expenses or in return on investment?

We have been searching for information about the importance of direct marketing mailings as proportion of all marketing activities in the Netherlands, but this information does not seem to be publicly available. We acknowledge that the investigated marketing materials are only a part of all marketing activities and sending marketing materials has become less common than it used to be (ref. Jeuck et al.). However, the amount of collected materials shows that pharmaceutical companies still invest in this type of marketing, indicating that it continues to be relevant. In the discussion, we added a short explanation about the relevance and importance of marketing materials related to all marketing activities (line 442).

Did the authors include material left behind by company sales representatives who visited doctors' offices?

The practices were instructed to collect materials sent by mail only. Although we cannot exclude that practices also collected other materials, this seems unlikely based on the clear instructions and the fact that most of the included materials could be identified as sent by mail afterwards, because of the presence of envelops and addresses. In addition, the visits of sales representatives has become less common in general practices in the last years . In the methods section, we further clarified the collecting instructions (line 148, 149 and 152). Also, we provided more details in the discussion about our focus on marketing materials sent by mail (line 425). 

Were the medical journals that general practitioners collected ones that they subscribed to or ones that were sent to them for free? Was there any difference in the type of persuasion strategy between the ads in the two different types of journals?

Based on the collected materials, we were not able to distinguish between journals with or without subscription. Therefore, we did not analyze differences between the journals. Also, the number of journals in the dataset is small, so that analyses would not be very meaningful 

In investigating the representativeness of the practices, did the authors collect data about whether general practitioners saw sales representatives and if so how often, e.g., visits per week? Seeing sales representatives may be a proxy for the influence that direct mailing might have on prescribing behaviour.

We agree that the inclusion of additional data, such as the possibility of sales representatives being a proxy for the influence on prescription behaviour, would have added further insights to our findings. However, we made a conscious decision not to include additional data that should be provided by healthcare professionals themselves, in order to minimize barriers to participation. Therefore, we only included publicly available information on the practice characteristics (line 138).

Since we acknowledge the importance of investigating the influence of marketing activities on prescription behaviour, we emphasized the importance of further research towards the influence of different marketing strategies in the discussion (line 409). 

In discussing educating practitioners about persuasion strategies the authors might want to refer to the following article: Mansfield et al. Educating health professionals about drug and device promotion: advocates’ recommendations. PLoS Medicine 2006;3:e451.

We thank the reviewer for this suggestion and added the reference (line 476). 

Reviewer #2: 1. Median year of launch of studied medicine was 2009 and the study conducted in June 2022. So, the word “new” may be omitted from the title further more in line 401 – 405 authors have mentioned that there is slow uptake of new medicine in Dutch practices.

We thank the reviewer for the useful feedback and suggestions on our manuscript. 

Based on this comment and the suggestion below to include a table with all introduction years, we carefully re-analyzed our data and discovered a mistake in the calculation of the median year of introduction. The median year of introduction of medicines is 2012 instead of 2009. We have corrected the median in the manuscript and apologize for this mistake. 

Although the median year was 2012 instead of 2009, we think the conclusion still stands, since it means that at least half of the medicines is already available for 10 years or longer. We therefore omitted the word ‘new’ from the title. Furthermore in the discussion (line 449), we clarified the statement about the slow uptake of new medicines in Dutch practices. 

2. Why the authors have calculated median year of launch? Justify your argument.

We calculated the median year of introduction to gain insight in the novelty of marketed medicines. This explanation was added in the methods (line 183). 

3. Provide of list of medicine with launch year.

This table has been added. Because of its size, we suggest to place this table in the appendix. 

4. 1966- first year of introduction, such products are called cash cow products which generally need no marketing so authors should provide pictures of marketing material of such products.

We think it is, due to privacy and marketing reasons, not desirable to share the pictures of marketing materials in the publication. Perhaps the newly inserted table, combining the names of the marketed products and the year of introduction will provide the desired insight in the ‘old’ products that are still being marketed. For reviewing purposes, we are willing to share pictures of all marketing materials with the reviewer and editor if desired. Please let us know if that is wanted, so we can provide access to the pictures. 

5. Dr. Cialdini is very well known for his work on “science of influence” but his work has also been criticized globally, so I suggest that authors shall give more references on the seven attributes of influence.

We acknowledge the criticism on the framework of Cialdini. However, numerous studies have used this framework and have proven its usefulness and effectiveness in obtaining insights in influencing intentions and behaviour. We have chosen for the principles of Cialdini because they are well-known and the model offers a useful framework for a first investigation towards persuasion strategies in marketing materials as our study aimed to do. We added both the criticism on the Cialdini framework and this explanation in the introduction (line 104). 

6. Emotional pressure is the eighth attribute for which references are required.

7. Authors should amend legend of Table I because in the legend they have stated seven strategies of influence by Cialdini but they have actually mentioned 8 strategies.

Please allow us to respond to comment #6 and #7 simultaneously. Table 1 mentions the seven principles Cialdini has described. The eight principle ‘emotional pressure’ was identified by our research team during the inductive analysis of persuasion strategies in marketing materials collected during a pilot. We therefore feel that it is more appropriate to limit the principles in the introduction and table 1 to those actually described by Cialdini and to clarify in the methods and results section the newly identified principle. We have described the deliberations leading to the addition of this eight mechanism in more detail in the methods (line 188) and discussion (line 381). In addition, we made a reference to previous studies towards the role of emotion in persuasion and marketing in the discussion (line 385). 

8. Authors have cited 41 articles out of which 31 references are older than 5 years so much so that one reference is from an article published in 1940. If use of old references is in conformity with journal's policy then it is ok.

The reference from 1940 is explicitly cited in the manuscript, to state that the use of marketing materials has been present for ages (line 433). This provides the context of marketing as a long-lasting phenomenon.

9. Reference number 29 is on “urbanization data” and should be with more detail in the reference list.

We provided more detail about this reference (line 568).

---

## [Decision Letter · Decision Letter 1]

14 Aug 2023

Marketing of medicines in primary care: an analysis of direct marketing mailings and advertisements

PONE-D-23-07120R1

Dear Dr. Marloes Dankers,

We’re pleased to inform you that your manuscript has been judged scientifically suitable for publication and will be formally accepted for publication once it meets all outstanding technical requirements.

Kind regards,

Naeem Mubarak, PhD

Academic Editor

PLOS ONE

Additional Editor Comments (optional):

The manuscript requires no further revisions and may be accepted for publication. 

Additional Author Comments (optional):

The manuscript requires no further revisions.  

Reviewers' comments:

Reviewer's Responses to Questions

**Comments to the Author**

1. If the authors have adequately addressed your comments raised in a previous round of review and you feel that this manuscript is now acceptable for publication, you may indicate that here to bypass the “Comments to the Author” section, enter your conflict of interest statement in the “Confidential to Editor” section, and submit your "Accept" recommendation.

Reviewer #1: (No Response)

Reviewer #2: All comments have been addressed

2. Is the manuscript technically sound, and do the data support the conclusions?

Reviewer #1: Yes

Reviewer #2: Yes

3. Has the statistical analysis been performed appropriately and rigorously? 

Reviewer #1: Yes

Reviewer #2: Yes

4. Have the authors made all data underlying the findings in their manuscript fully available?

Reviewer #1: Yes

Reviewer #2: Yes

5. Is the manuscript presented in an intelligible fashion and written in standard English?

Reviewer #1: Yes

Reviewer #2: Yes

6. Review Comments to the Author

Reviewer #1: The revisions that the authors have made have successfully dealt with my original concerns. There are now a few relatively minor issues that need to be resolved.

1. At points, the English is somewhat awkward and the manuscript should be thoroughly reviewed by someone who has English as a first language.

2. Line 137: The authors should provide a bit of detail about dispensing status, e.g., is this something that any GP could do, is it only done in rural areas, are there any qualifications necessary to be able to do it, etc.

3. Line 274: When the authors say "relatively small" presumably they mean of limited monetary value but this meaning should be explicitly stated.

Reviewer #2: The authors have improved the manuscript as reviewed. Title has been revised. Authors have explained about calculation of median year and its reason. They have explained Cialdini's framework and emotional pressure fo persuassion. Authors have revised and corrected the reference list.

7. PLOS authors have the option to publish the peer review history of their article (what does this mean?). If published, this will include your full peer review and any attached files.

Reviewer #1: **Yes: **Joel Lexchin

Reviewer #2: No

---

## [Editor Report · Acceptance letter]

18 Aug 2023

PONE-D-23-07120R1 

Marketing of medicines in primary care: an analysis of direct marketing mailings and advertisements 

Dear Dr. Dankers:

I'm pleased to inform you that your manuscript has been deemed suitable for publication in PLOS ONE. Congratulations! Your manuscript is now with our production department. 

Kind regards, 

on behalf of

Dr Naeem Mubarak 

Academic Editor

PLOS ONE